



# Deformation lines in Arctic sea ice: intersection angles distribution and mechanical properties

Damien Ringeisen[1,2,3,*], Nils Hutter[4,1,*], and Luisa von Albedyll[1]

[1]Alfred-Wegener-Institut, Helmholtz-Zentrum für Polar- und Meeresforschung, Bremerhaven, Germany
[2]MARUM – Center for Marine Environmental Sciences, Leobener Str. 8, 28359, Bremen, Germany
[3]Department of Atmospheric and Oceanic Sciences, McGill University, Montréal, QC, Canada
[4]Cooperative Institute for Climate, Ocean, and Ecosystem Studies (CICOES), University of Washington, Seattle, WA, United States
[*]These authors contributed equally to this work.

**Correspondence:** Damien Ringeisen (damien.ringeisen@mcgill.ca) and Nils Hutter (nhutter@uw.edu)

**Abstract.** In Arctic sea ice, the intersection angles between Linear Kinematic Features (LKFs) are linked to the internal mechanical properties. Sea ice rheological models struggle to reproduce the intersection angles between LKFs in Arctic sea ice. We aim to obtain an intersection angle distribution (IAD) from observational data to serve as a reference for high-resolution sea ice models and to infer the mechanical properties of the sea ice cover. We use the sea ice vorticity to discriminate between acute and obtuse LKFs intersection angles within two sea ice deformation datasets: the RGPS and a new dataset from the MOSAiC drift experiment. Acute angles dominate the IAD, with single peaks at $48° \pm 2$ and $45° \pm 7$. The IAD agrees well between both datasets, despite the difference in scale, time periods, and geographical location. The divergence and shear rates of the LKFs also have the same distribution. The dilatancy angle (the ratio of shear and divergence) is not correlated with the intersection angle. Using the IAD, we infer an internal angle of friction in sea ice of $\mu_I = 0.66 \pm 0.02$ and $\mu_I = 0.75 \pm 0.05$. The shape of the yield curve or the plastic potential derived from the observed IAD resembles the teardrop or a Mohr–Coulomb shape. With those new insights, sea ice rheologies used in models can be adapted or re-designed to improve the representation of sea-ice dynamics.

## 1 Introduction

The deformation patterns of sea ice in the Arctic Ocean are dominated by narrow lines where deformation concentrates (Schall and van Hecke, 2010), known as Linear Kinematic Features (LKFs), failure lines or shear bands (Kwok, 2001). LKFs play a primary role in the mass and energy budget of the Arctic ocean. First, the creation of thicker ice (ridges) or open water (leads), where sea ice growth is enhanced in winter, takes place along LKFs (Stern et al., 1995; Hopkins, 1994; von Albedyll et al., 2020, 2022). Second, shear motion and brine injection from sea-ice growth along LKFs influence the halocline (McPhee et al., 2005; Itkin et al., 2015; Nguyen et al., 2012). Finally, open leads govern the polar ocean-atmosphere heat and moisture exchange despite their small total area (Maykut, 1978; Untersteiner, 1961; Tetzlaff et al., 2015). Thus, LKFs are an essential



component of the Arctic climate system and need to be accurately represented in regional models for reliable regional weather predictions, navigation charts, and services to Arctic communities.

LKFs emerge from the mechanical properties of sea ice. Sea ice is often described as a granular material (Overland et al., 1998; Tremblay and Mysak, 1997), which exhibits brittle properties (Schulson, 2002; Dansereau et al., 2016). Important me-
chanical properties of sea ice are imprinted in the orientation of the failure lines relative to the stress direction: Two mechanical parameters are known to play a role in the orientation of the failure lines relative to the stress direction in granular materials (Vermeer, 1990): (1) the material's strength threshold to internal stress leading to deformation, especially the ratio of shear strength to compression strength, named the internal angle of friction (Coulomb, 1773), and (2) the dilatancy, or motion perpendicular to the slip line (Roscoe, 1970). The sea ice motion of (2) can be observed via remote sensing. Still, the internal stress
magnitude and direction of (1) cannot be observed at the Arctic scale, such that it is impossible to measure the orientation of the LKFs with respect to the stress direction. Instead, we use the vorticity at the intersections of LKFs to infer the main stress direction and link it to the intersection angles of the LKFs.

Former studies report single intersection angles based on small sample sizes across large spatial scales (100m-100km), for example, $28°$ (Marko and Thomson, 1977), $30 \pm 4°$ (Erlingsson, 1988), $30°$ (Walter and Overland, 1993), and $34°$ to
$36°$ (Cunningham et al., 1994). From the RADARSAT Geophysical Processor System (RGPS) sea ice motion dataset (Kwok et al., 1998), sea ice deformation data were obtained over the Arctic Ocean with a period of 3 days, allowing the automated extraction of LKFs locations and angles (Hutter et al., 2019; Linow and Dierking, 2017). Recent work based on an LKFs tracking algorithm report an intersection angles distribution (IAD) between $0°$ and $90°$ with a peak between $40°$ to $50°$ (Hutter and Losch, 2020). Multi-scale directional analysis (MDA) on the RGPS dataset also shows that small intersection angles are
dominant (Mohammadi-Aragh et al., 2020).

None of the current sea ice models can reproduce the observed distribution of LKF intersection angles (Hutter et al., 2022). This hints that the current implementations of sea ice rheological models are not accurate enough to describe the mechanical properties of sea ice. Most climate models today simulate sea ice as Viscous-Plastic (VP) medium with an elliptical yield curve and normal flow rule (Hibler, 1979; Stroeve et al., 2014). Diffuse small deformations are represented by viscous behavior,
while the large deformations along LKFs are represented by plastic behavior (Hutchings et al., 2005). High-resolution sea-ice VP models can represent LKFs at scales 5-7 times larger than their horizontal grid spacing (Hutter et al., 2018; Bouchat et al., 2022; Hutter et al., 2022). The intersection angle of LKFs depends in the VP rheology framework on parameters that define the constitutive equation: the yield curve that defines the stress at failure and the plastic potential that defines the post-failure deformation called the flow rule (Ringeisen et al., 2021, 2019). Based on their observations of LKFs intersection
angles, Erlingsson (1991) proposed an internal angle of friction of $phi = 15° \pm 2°$, while Marko and Thomson (1977) proposed $\phi \simeq 62°$. Using a small set of LKF intersecting angles and the assumption that the major principal direction of the sea ice internal stress is perpendicular to the wind direction, Wang (2007) proposed the *curved diamond* yield curve. However, there seems to be a need for improvement. LKFs-tracking algorithms show that the current VP models overestimate the intersection angles, with an IAD peaking at $90°$ (Hutter et al., 2019; Hutter and Losch, 2020). This behavior is shared by all other rheological
models as a recent comparison of state-of-the-art models revealed (Hutter et al., 2022), and is also observed using MDA





(Mohammadi-Aragh et al., 2020). To improve the IAD in high-resolution sea ice models, the observed IAD could be used to improve the definition of weakly constrained sea ice rheological parameters: the yield curve and the plastic potential.

Studying the intersection angles can provide important insights into two key questions: First, is there a relationship between intersection angles and the divergence (opening or ridging) along the LKF? In other words, does the hypothesis of the normal flow rule, as it is currently used in sea ice VP models, hold (Ringeisen et al., 2019, 2021)? Second, does the observed IAD allows us to deduce the mechanical properties of sea ice and thereby constrain the shape of the yield curve or the plastic potential? To answer these questions, we see the need to revisit the IAD as it is presented in the literature. First, the angles reported in previous studies are given in the interval between 0 and 90°, leaving undefined if these angles are acute (between 0 and 90°) or obtuse (90 to 180°) compared to the principal stress direction. Both cases need to be separated as they are linked to different slopes of the yield curve/plastic potential, hence to different shapes of the yield curve/plastic potential (Ringeisen et al., 2019). Second, we need to filter only conjugate pairs of LKFs, i.e., intersecting LKFs that formed simultaneously under compressive forcing, which is challenging given non-continuous coverage with satellite observations.

In this paper, we use satellite-derived sea-ice drift and deformation to address the gaps outlined above. Deformation concentrates along the LKFs, and vorticity identifies the LKFs formed under compressive force. Tracking of the LKFs allows for identifying those that formed simultaneously. Therefore, we can distinguish between conjugate and non-conjugate intersection angles and discriminate between conjugate obtuse and acute intersection angles. We apply this method to the RGPS dataset and new high-resolution deformation data surrounding the 2019/2020 MOSAiC expedition. Both datasets have different coverage and resolution; thus, they indicate if intersection angles vary in the ice cover depending on the spatial scale and geographical location in the Arctic. We aim to obtain an IAD as a reference for high-resolution sea ice models and to infer the mechanical properties of the sea ice cover, e.g., the yield condition and/or the plastic potential.

The remainder of this paper is structured as follows: Section 2 presents the different datasets used in this study (Section 2.1) and the algorithm for the measurements of the angles between 0 and 180° (Section 2.2). Section 3 presents the results of the intersection angles for the different datasets, the divergence along LKFs, seasonal variations, estimations of internal angles of friction, and an estimation of the shape of a yield curve for sea ice modelling. Discussion and Conclusions follow in Section 4 and 5.

## 2 Methods and data

### 2.1 Datasets

In this study, we will use two satellite-based sea-ice drift datasets from which sea-ice deformation is derived. Thanks to the synthetic aperture radar (SAR) data from which the drift is calculated, the datasets are available independent of weather conditions and during the polar night. The high spatial resolution (1.4 km. and 12.5 km) of the deformation datasets enables us to identify individual LKFs.





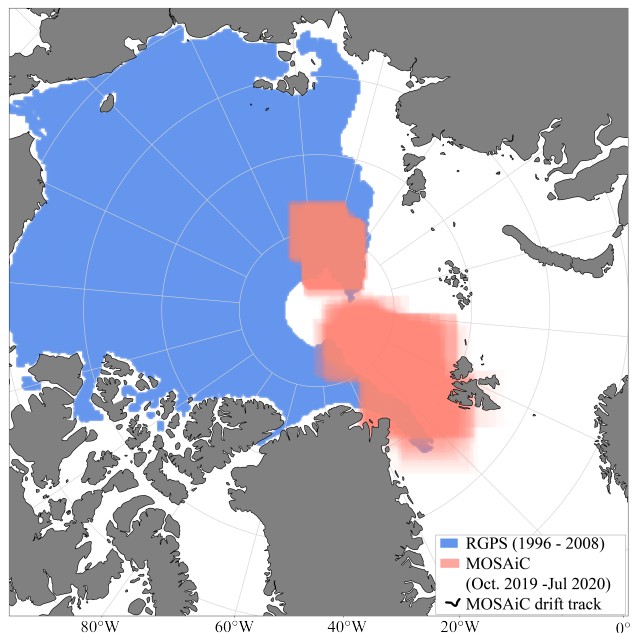

**Figure 1.** Coverage of the RGPS and MOSAiC LKF datasets

### 2.1.1 RADARSAT Geophysical Processor System

RGPS is a widely used drift and deformation dataset based on RADARSAT SAR images (Kwok, 1998). The dataset covers the Amerasian Basin of the Arctic Ocean (Fig. 1) for twelve winters from 1996 to 2008. Sea ice drift is derived by tracking points that are spaced 10 km apart in SAR images. Deformation rates are computed from these Lagrangian drift paths and are interpolated to a regular 12.5 km grid. Hutter et al. (2019) applied detection and tracking algorithms to the regular gridded data set and extracted deformation features, which are publicly available (Hutter et al., 2019) and are closer analyzed in this study.

### 2.1.2 Sentinel (MOSAiC)

In addition, we compute ice drift and deformation based on Sentinel-1 SAR scenes (von Albedyll and Hutter, submitted). We base this dataset on HH-polarized scenes with a spatial resolution of 50 m. The scenes are located along the drift of the Multidisciplinary drifting Observatory for the Study of Arctic Climate (MOSAiC) expedition (Nicolaus et al., 2022) from October 5, 2019, to July 14, 2020, except for the period between January 14 and March 15, when the ship was north of the satellite coverage (Figure 1). Typically, the time between two scenes was one day, with a few exceptions of 2-3 days, and the size of the scenes was on average 200×200 km. We compute ice drift fields based on a pattern-matching ice tracking algorithm introduced by Thomas et al. (2008, 2011) with substantial modifications by Hollands and Dierking (2011). We retrieve divergence, shear, total deformation, and vorticity from the regularly gridded sea-ice drift output at 1.4 km resolution following the approach described in von Albedyll et al. (2020) and Krumpen et al. (2021b).





## 2.2 LKF detection and Angles measurement

### 2.2.1 LKF detection

We use the algorithms presented in Hutter et al. (2019) to detect and track LKFs in both deformation data sets. LKFs are detected in four steps: (1) pixels are marked as LKF if their deformation rates exceed the average deformation rate of the neighboring pixels, (2) all LKFs in the binary mask of LKF pixels are reduced to their skeleton using morphological thinning, (3) the binary map is divided into smallest possible LKF segments, and (4) segments are reconnected to one LKF based on the probability of them belonging to the same LKF that is computed from their distance, orientation, and deformation rate

magnitude differences. Next, the drift data is used to advect LKFs and track them over time. Note that to exclude a direct influence of the coast, those regions are excluded from the RGPS dataset, while the MOSAiC dataset only covers pack ice (see Fig. 1).

For RGPS, we use the publicly available LKF data (Hutter et al., 2019) using the original version of the code (Hutter, 2019). For the higher resolution MOSAiC Sentinel data, we add two modifications to the original version of the detection code: (1) We

apply a directional filter to the input deformation rates to reduce grid-scale noise. The directional filter is a 1-d kernel spanning 7 pixels that is applied along the direction of lowest variability at each pixel to reduce noise but still preserve the linear structure of LKFs in the deformation data. (2) The morphological thinning routine was modified to align the LKF skeletons in the binary maps to the position of the highest deformation rates across the LKF.

### 2.2.2 Angles measurements

In both LKF datasets, pairs of LKF that intersect and are formed within the same time step are extracted, and the angle of intersection is measured following the approach of Hutter et al. (2019) and Hutter et al. (2022). To differentiate between intersection angle that are acute ($< 90°$) or obtuse ($> 90°$), we use the vorticity $\left(\dot{\epsilon}_{\text{vort}} = \frac{1}{2}\left(\frac{\partial u_1}{\partial x_2} - \frac{\partial u_2}{\partial x_1}\right)\right)$ along the LKFs as shown on Figure 2. From the vorticity information, we separate the data set into two categories: The conjugate angles (acute and obtuse together) and the non-conjugate angles. For conjugate angles, the major stress direction can be identified from

the ice motion indicated by the opposite sign vorticity along the intersecting LKFs (Fig. 2). For equal sign vorticity along both LKFs, the ice motion field does not allow identifying the main stress direction, and we classify the intersecting pair as non-conjugate.

While the generation of conjugate faults is explained by the failure of ice under compressive loading, the reasons for the existence of non-conjugate failure are less obvious. We show the distribution of the non-conjugate angles and explore reasons

for these non-conjugate faults in Appendix A.

## 3 Results

In the following sections, we present the results of our investigation of intersection angles in both MOSAiC and RGPS data sets. We compare the intersection angles distribution (IAD), the relationship between angles and dilatancy, and how these





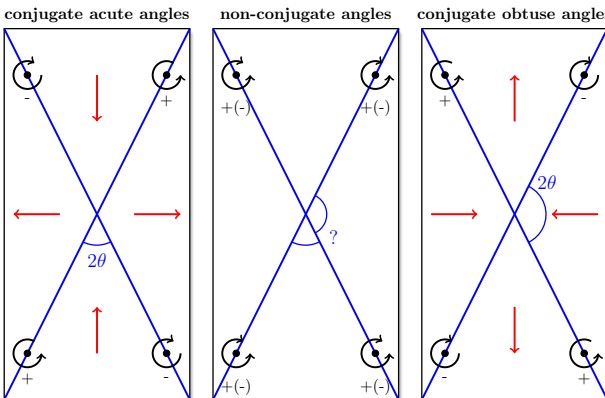

**Figure 2.** Schematics showing the difference between conjugate failure lines with acute and obtuse angles, and with non-conjugate failure lines. The vorticity of the drift allows making the difference between conjugate and non-conjugate failure lines.

distributions inform us about sea ice dynamics, especially the internal angle of friction, and the possible shape of the yield curve or the plastic potential for sea ice VP models.

### 3.1 Example of LKFs intersection

Figure 3 presents three examples of LKFs intersections from the MOSAiC data set, one conjugate acute angle (Fig. 3 a), one conjugate obtuse angle (Fig. 3 c), and one non-conjugate angle (Fig. 3 b). Especially for conjugate acute angles, we observe that LKFs with the same orientation have vorticities of the same sign. These patterns agree with the concept of fracture in diamond shapes as shown in DEM simulation (Wilchinsky et al., 2010) or in theoretical works (Pritchard, 1988). For the non-conjugate and the obtuse angles, the fracturing pattern is more chaotic, a possible effect of heterogeneities in the ice strength, e.g., in the ice thickness field or rapidly changing forcing fields.

In the following, we focus on intersection angles between conjugate LKFs, which we can classify as acute or obtuse. We include only LKFs that formed during the same time step of observation.

### 3.2 Intersection angles

Figure 4 shows the probability density function (PDF) of the intersection angle of conjugate LKFs for the MOSAiC and RGPS datasets. The PDF of both datasets agrees remarkably well. Both IAD peak at acute angles, with a modal value in the range between $40°$ and $50°$. Also, we find few large angles. For both datasets, around 80% of the conjugate angles are acute with around 25% of the conjugate angles ranging between $30°$ and $60°$.

We fit both IAD to an exponentially modified Gaussian (exGaussian) distribution with a Maximum Likelihood Estimator (MLE). The parameters of the fit are given in the caption of Fig. 4. The formula of the exGaussian distribution is,

$$f(x; \mu, \sigma, \tau) = \frac{1}{2\tau} e^{\frac{1}{2\tau}\left(2\mu + \frac{\sigma^2}{\tau} - 2x\right)} \operatorname{erfc}\left(\frac{\mu + \frac{\sigma^2}{\tau} - x}{\sqrt{2}\sigma}\right), \tag{1}$$





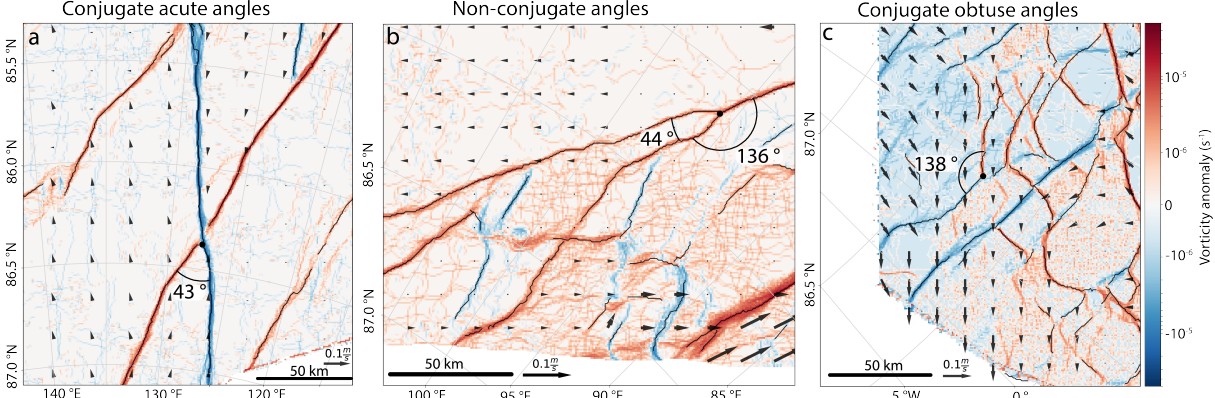

**Figure 3.** Example of LKFs intersections with the vorticity anomaly and sea ice drift within the MOSAiC dataset. Panel **(a)** shows an intersection with an acute angle (Jan 1-2, 2020), panel **(b)** shows an intersection with a non-conjugate angle (Nov 11-12, 2019), and panel **(c)** shows an intersection with an obtuse angle (Mar 15-16, 2020). The arrows indicate the velocity anomaly of the sea ice drift calculated for the displayed data. Detected LKFs are plotted as thin black lines. The colorbar of the vorticity anomaly is the same for all panels.

with

$$\mathrm{erfc}(x) = \frac{2}{\sqrt{\pi}} \int\limits_x^\infty e^{-t^2} \, dt. \tag{2}$$

The goodness of the fit is tested with a Monte-Carlo test with $10\,000$ iterations taking into account the discrete nature of intersection angles between LKFs that are defined on a regular grid (Clauset et al., 2009; Hutter et al., 2019). We also tested the Logarithmic Normal Distribution and the Skew Normal Distribution, but these fits failed the Monte-Carlo test. The fitted exGaussian distributions show a modal peak at $49° \pm 1°$ for the RGPS dataset and $42° \pm 4°$ for the MOSAiC dataset (Figure 4). Considering the different spatial and temporal resolutions as well as the different ice regimes sampled, this agreement is 160 remarkable and allows us to generalize conclusions on sea ice properties.

The PDF of intersection angles does not vary seasonally for the RGPS dataset. For the MOSAiC dataset, there are too few intersection angles to study seasonal variations (not shown). Appendix A presents and discusses the PDF of non-conjugate intersection angles.

### 3.3 Divergence and convergence along leads

Besides the IAD, both the MOSAiC and the RPGS dataset agree in the shape of the distribution of the divergence, shear, and total deformation rates along LKFs (Fig. 5a, b, and c, respectively). Since RGPS and MOSAiC differ in spatial resolution by one order of magnitude and deformation rates are known to be scale-dependent, we normalize the divergence and shear to compare the relative frequencies. On average, we find more divergent ice motion along LKFs in the MOSAiC data set compared to the RGPS data set (Fig. 5b), which can be explained by the divergent regime in the Transpolar drift compared to 170 the Beaufort Gyre that features more compressive settings (Fig. 1). The RGPS data set shows higher shear deformation that





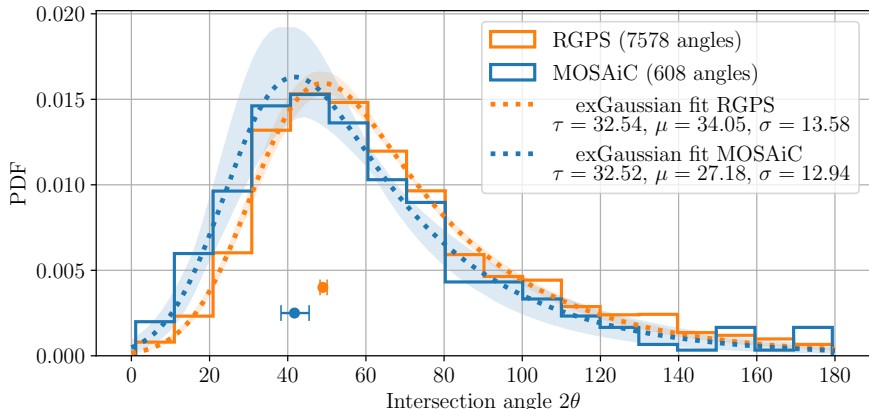

**Figure 4.** Probability density function of the conjugate intersection angles for the MOSAiC (blue) and RGPS (orange) datasets. The dashed lines show the MLE fit to an exponentially modified Gaussian (exGaussian) distribution (Eq. 1), where the fitted distribution parameters are shown in the legend. The shading shows the 1-$\sigma$ error of the distribution fits. We show the position of the distribution's peak with its 1-$\sigma$ error: $49° \pm 1°$ for the RGPS dataset and $42° \pm 4°$ for the MOSAiC dataset.

originates from the circular motion of the Beaufort Gyre. The higher shear rates also result in higher total deformation rates in the RGPS data set. We can calculate the dilatancy angle along the LKF, $\delta$, i.e., the angle of deformation defined by the ratio of shear and divergence $\tan(\delta) = \frac{\dot{\epsilon}_{\text{div}}}{\dot{\epsilon}_{\text{shear}}}$ (Tremblay and Mysak, 1997). The presence of smaller dilatancy angles in the distribution (Fig. 5d) shows well that divergence is more frequently occurring in the MOSAiC dataset (Fig. 5d).

Further, we analyze the relationship between intersection angles and dilatancy angles (Figure 6). We find a weak correlation between the intersection angle and the dilatancy angle, with a correlation coefficient $\rho = -0.5$ for MOSAiC, and a very weak correlation $\rho = -0.2$ for RGPS. The theory of Roscoe angles $\theta_R$ (Roscoe, 1970) states that the dilatancy (that is, the orientation of the flow rule) controls the orientation of the LKFs. In doing so, intersection angles $2\theta$ can be described as a function of the dilatancy angle $\delta$ and vice versa by

$$2\theta = \arccos(\tan(\delta)) \tag{3}$$

shown as a red dashed line in Figure 6). In contrast to theory, we find that both are not linked following this functional form (Figure 6), showing only weak correlations ($\rho_R = 0.46$ for MOSAiC and $\rho_R = 0.20$ for RGPS) and even negative determination coefficients $R^2$, meaning that a constant line would be a better fit than Eq. (3).

     These findings contradict the concept of the Roscoe Angle $\theta_R$ and the idea of a normal flow rule, as we observe only
weak correlations between intersection angles and dilatancy angles. We, however, note that the MOSAiC dataset holds a higher correlation than the RGPS dataset. The non-correlations could arise from an observational bias due to the low temporal resolution. Increasing the temporal resolution might show a correlation as it would resolve the deformation just after failure. Finally, we note that our observations confirm the theoretically expected range of the dilatancy angle: As expected from





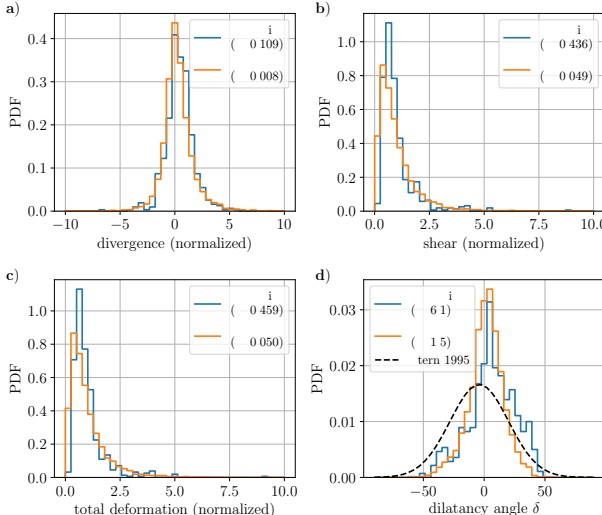

**Figure 5.** Probability distribution function of normalized convergence **(a)**, normalized divergence **(b)**, and normalized total deformation **(c)** along failure lines, as well as the dilatancy angle $\delta = \tan^{-1}\left(\frac{\dot{\epsilon}_{\mathrm{div}}}{\dot{\epsilon}_{\mathrm{shear}}}\right)$ **(d)**. The means used for the normalization are given in parentheses in the legend. The black curve on panel **(d)** shows the Gaussian curve with the parameters presented in Stern et al. (1995, their Fig. 4) for comparison.

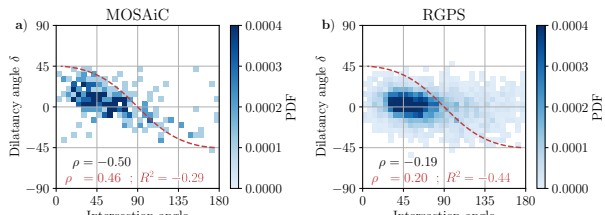

**Figure 6.** Scatter plot of the dilatancy angles $\delta$ versus the intersection angles between LKFs. The correlation between the intersection angle and the dilatancy angle is $\rho = -0.5$ for MOSAiC and $\rho = -0.19$ for RGPS. The red dashed line shows the relationship between dilatancy and angles expected following a normal flow rule or the Roscoe angle theory. The correlation between this prediction and the observed dilatancy angles are $\rho_R = 0.46$ for MOSAiC and $\rho_R = 0.20$ for RGPS. All correlations are significant.





Roscoe's theory and the normal flow rule condition, dilatancy angles lower than $45°$ and above $135°$ are very rare in our
observations (Fig. 6 and Fig. 5d).

### 3.4 Mechanical properties of sea ice

#### 3.4.1 Estimation of the internal angle of friction

The peak of the IAD shows a preferred angle of failure of sea ice that can be used to estimate the internal angle of friction
of sea ice within the framework of the Mohr–Coulomb failure criterion (Coulomb, 1776). The internal angle of friction is the
ratio of shear stress $\tau$ and normal stress $\sigma$ at which the material yields.

The internal angle of friction is given by $\phi = \frac{\pi}{2} - 2\theta$, where $2\theta$ is the peak of the IAD. Within the Coulombic framework,
the internal angle of friction is linked to a Coulombic shear criterion, such as

$$\tau = \mu\sigma + \tau_0 \tag{4}$$

with $\mu = \tan(\phi)$. Similarly, the criterion can be translated in invariant stress space ($\sigma_\mathrm{I}$,$\sigma_\mathrm{II}$) for the construction of a yield curve
(Ringeisen et al., 2019) and is given by

$$\sigma_\mathrm{II} = \mu_\mathrm{I}\sigma_\mathrm{I} + c_\mathrm{I}. \tag{5}$$

with $\mu_\mathrm{I} = \sin(\phi)$.

Using these formulas with the observed IAD from our study, we find:

- The peak intersection angles for the RGPS dataset of $2\theta = 49° \pm 1°$ implies an internal angle of friction of $\phi = 41° \pm 1°$,
$\mu = 0.87 \pm 0.03$, and $\mu_\mathrm{I} = 0.66 \pm 0.02$.

- For the MOSAiC dataset, the intersecting angle of $2\theta = 42° \pm 4°$, we get $\phi = 48° \pm 4°$, $\mu = 1.12 \pm 0.15$, and $\mu_I = 0.75 \pm 0.05$.

(By following the framework of Erlingsson (1988, 1991), a breaking index of $i = 2$ gives internal angles of friction of $\phi \simeq 25°$
and $\phi \simeq 65°$.)
Note that for this calculation, we only take the peak of the IAD into account, neglecting the presence of other intersection
angles in the PDF.

#### 3.4.2 Estimation of the shape of a yield curve or a plastic potential

Instead of using only a single angle to derive the mechanical properties of sea ice, we can also use the complete PDF of the
intersection angles (Figure 4) to create an approximation of the shape of the yield curve or plastic potential within the viscous-
plastic framework. Below, we consider that the curve that we derive can be a yield curve or a plastic potential because it is
still an open question of which of these mechanical properties set the intersection angles of LKFs in sea ice plastic models.
There are indications that the plastic potential could be responsible (Ringeisen et al., 2021). To reconstruct these curves from





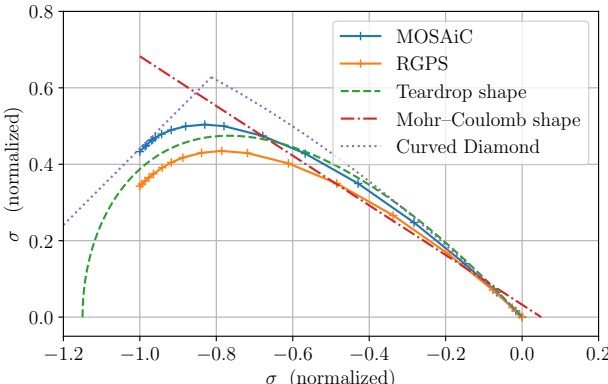

**Figure 7.** Yield curve/Plastic potential constructed from the PDF of the intersection angles. The red dash-dotted line, the green dashed line, and the dotted violet line show a Mohr–Coulomb yield curve (Tremblay and Mysak, 1997), a teardrop shape (Zhang and Rothrock, 2005; Ringeisen et al., 2022), and a curved diamond (Wang, 2007), respectively, for comparison. For the Mohr–Coulomb yield curve, the slope is $\mu = 0.75$, as derived in Sec. 3.4.1

the IAD, we follow the results of (Wang, 2007), Ringeisen et al. (2019), or Ringeisen et al. (2021). The slope of different parts of the yield curve (Coulomb angle) or plastic potential (Roscoe angle) are linked to different intersection angles:

$$2\theta = \arccos\left(-\frac{\partial F_{\mathrm{I}}}{\partial \sigma_{\mathrm{I}}}\right) \tag{6}$$

where the function $F_{\mathrm{I}} = \sigma_{\mathrm{II}}(\sigma_{\mathrm{I}})$ defines the yield curve or the plastic potential in the invariant stress space ($\sigma_{\mathrm{I}}, \sigma_{\mathrm{II}}$). In the following, we make the hypothesis that the number of intersecting LKFs within a bin of angles is proportional to the length of the yield curve/plastic potential curve that creates this angle. Here, our underlying hypothesis is that all the points on the yield curve/plastic potential are equally likely. An additional constraint is that the curve is required to be convex to agree with the convexity condition of Drucker's postulate of stability (Drucker, 1959).

For each bin of angles in the observed PDF, we compute a segment of the yield curve (or plastic potential) that has the length of the PDF value and the slope given by the angle in the center of this bin from Eq. (6). We start the construction of the curve at the origin of the invariant stress. We start from the smallest angles and iterate through the PDF, with the start point of each segment being the tip of the previous segment. Note that as long as the intersection angles are either monotonically increasing or decreasing, our method necessarily leads to a convex yield curve. Finally, the curve values are normalized to have the tip at $\sigma_{\mathrm{I}} = -1$. Figure 7 shows the resulting shape.

The estimation of the obtained curves for the RGPS and MOSAiC dataset resemble a teardrop yield curve (Zhang and Rothrock, 2005; Rothrock, 1975; Ringeisen et al., 2022), a Mohr–Coulomb yield curve (Ip et al., 1991), or (to a lesser extent) the curved diamond yield curve (Wang, 2007), but clearly deviate from the elliptical yield curve (see Appendix B). For comparison, we added a teardrop, a Mohr–Coulomb, and a curved diamond yield curve in Fig. 7.





Note that the starting point of our curve is arbitrarily placed at the origin, as we did not consider tensile strength. However, adding tensile strength would not change the shape of the curve, which is central, but only the actual values of $\sigma_\mathrm{I}$, $\sigma_\mathrm{II}$. In other words, it is the shape of the curve that this important, and not its position in the ($\sigma_\mathrm{I}$, $\sigma_\mathrm{II}$) space.

In Appendix B, we present a proof-of-concept of this method. Using the IAD measured in three 2 km simulations with the
Massachusetts Institute of Technology general circulation model (MITgcm, Hutter et al., 2022) we show that the yield curves reconstructed with the presented method agree well with the elliptical yield curves used in the simulations.

## 4   Discussion

We show the intersection angle distribution (IAD) of conjugate faults in the Arctic sea ice during faulting events. The IAD shows the predominance of small intersection angles of $30°$ to $60°$. The predominance of small angles agrees with the previous
observations of intersection angles, which report angles between $28°$ and $36°$ (Erlingsson, 1988; Marko and Thomson, 1977; Walter and Overland, 1993; Cunningham et al., 1994). However, our observations show that also wider angles, and even obtuse angles, are present in sea ice, although less frequently. The spread and shape of the IAD can be explained by the presence of heterogeneities in the ice (open or refrozen leads, ridges, polynyas) that influence the orientation of LKFsWilchinsky and Feltham (2011), or by variations of internal confining pressure (Golding et al., 2010; Schulson et al., 2006). The heterogeneities
serve as the preferred direction of ice failure and, therefore, can alter the intersection angle from the single angles of Mohr–Coulomb theory. We note that the IADs of the RGPS and MOSAiC datasets look very similar in shape. Given the different scales, times, and regions of the datasets, we conclude that the shape of the IAD seems to be a characteristic of sea ice. We wonder how the shape of the IAD is linked to other typical characteristics of sea ice, like the floe size distribution (Rothrock and Thorndike, 1984; Stern et al., 2018) and the ice thickness distribution (von Albedyll et al., 2022; Thorndike et al., 1975).
Using the peak of the IAD, we made an estimation of the internal angle of friction from the Mohr–Coulomb's framework. Our estimates of $\mu_I = 0.66$ and $\mu = 0.75$ agree well with previous estimates of the internal angle of friction $\mu \in [0.6, 0.8]$ (Schulson et al., 2006). More importantly, we used the IAD to derive an approximation of a yield curve/plastic potential for sea ice VP models. Wang (2007) used a similar method to create the Curved Diamond yield curve, but with fewer angle observations and the strong assumption about inferring the unknown stress direction from coastal geometry. Using the along-
lead vorticity, we do not need to make this assumption. The shape resulting from our analysis is similar to a teardrop shape (Rothrock, 1975; Zhang and Rothrock, 2005; Ringeisen et al., 2022), a Mohr–Coulomb shape (Ip et al., 1991), or to a lesser extent, a Curved Diamond (Wang, 2007). In contrast, the shape we obtain does not fit the elliptical shape (Hibler, 1979) or the Parabolic Lens shape (Zhang and Rothrock, 2005). These findings are of great relevance for designing new rheologies for sea ice models, and the observed IAD can be used as a metric to assess the models' capability to represent LKFs.
In VP models, the orientation of the LKFs is tightly linked to the flow rule, i.e., the dilatancy or post-failure deformation, at least for the elliptical yield curve and plastic potentials Ringeisen et al. (2021). The observations presented here show no strong relationship between the observed dilatancy angle and the observed intersection angle, nor between the dilatancy angles and the expected angles from Roscoe theory (Roscoe, 1970) or the normal flow rule. We consider insufficient observations





or a general flaw in the VP rheological framework as potential reasons for this misfit between theory and observations. First, the temporal resolution of our observations might be insufficient to resolve double sliding cycles with positive and negative dilatancy (Balendran and Nemat-Nasser, 1993). In that case, we would not see the immediate post-fracture deformation, but the sliding of ice packs that alternate between dilatation and compression. This would lead to a random distribution of dilatancy angles and, thus, a de-correlation. We suggest further observational studies with a higher temporal resolution to confirm the de-correlation. Second, if confirmed by observations at higher temporal resolution, uncoupling between dilatancy and intersection angles would mean that the intersection angle is not influenced by the velocity characteristics of the medium. A confirmed uncoupling would question the capacity of the VP rheological framework to reproduce both the dilatancy and the LKFs intersection angles simultaneously.

The presented results of the IAD are robust in scale, resolution, and geographic area. The RGPS dataset has low spatial and temporal resolution but a large spatial and time coverage, whereas the MOSAiC dataset has higher resolution, but only covers the track of the MOSAiC drift experiment. The scale independence of the intersection angles agrees with the self-similarity and scaling properties of sea ice deformation (Marsan et al., 2004; Rampal et al., 2008; Hutchings et al., 2011; Weiss, 2013; Bouchat and Tremblay, 2020). While the LKFs at the kilometer scale resolution that we analyzed in this study are mostly systems of smaller-scale leads, the question remains if the observed deformation characteristics and IAD are still present at the floe scale ($< 100\,m$). Observational scaling studies analyzing sea ice deformation derived from ship-radar have shown that sea ice deformation follows the same scaling behavior down to scales of $50\,m$ (Oikkonen et al., 2017). This and laboratory experiments at sub-meter scale (Schulson et al., 2006; Weiss and Dansereau, 2017) indicate similar failure behavior across scales and, therefore, no scale-break. However, scaling characteristics are known to be only weakly linked to the representation of LKF intersection angles (Hutter et al., 2022) and therefore are potentially a poor proxy for the floe-scale IAD. Deformation derived from the shipborne ice radar from the MOSAiC drift experiment could bridge the gap in the scale analysis of the LKFs intersection angle and give insight into small-scale processes involved (rotation, reopening of leads) thanks to its high-temporal ($2\,sec$ - $10\,min$) and spatial ($10\,m$) resolution in the proximity (approx. $9\,km$) of the ship (Krumpen et al., 2021a).

## 5 Conclusions

Using the vorticity in sea ice deformation, we show that we can separate obtuse and acute intersection angles between sea ice Linear Kinematic Features (LKFs). Using this technique, we can now extract the Probability Density Function (PDF) of the intersection angles between 0 and 180°, instead of being limited to the range between 0 and 90°. We investigate intersection angles within two different deformation datasets: the RADARSAT RGPS product (Hutter et al., 2019), and the MOSAiC dataset from Sentinel-1 A/B (von Albedyll and Hutter, submitted).

The PDFs of intersection angles show that acute angles dominate in both datasets, with PDFs peaking at 48°(RGPS) and 45°(MOSAiC). Both PDFs are described by an exponentially modified Gaussian that agrees remarkably in shape, i.e., the Intersection Angle Distribution (IAD) is scale invariant. The distributions of divergence and shear rates along the LKFs also agree when taking the scaling of deformation rates into account. Both indicate scale-invariant behavior of fracture mechanics





and intersection angles that remain to be tested at the floe scale. We do not find a relationship between the dilatancy angles along the leads and their corresponding intersection angles, which could be an artifact of the temporal resolution of a minimum of one day that might "smear" the deformation during instantaneous failure with post-fracture sliding. If not falsified with deformation

data at very high temporal resolution, e.g., ship radar, the de-correlation of dilatancy and intersection angles contradicts the normal flow rule assumption used within the VP framework.

We infer the mechanical properties of sea ice from the observed IAD. Following methods from previous papers, we estimate the internal angle of friction at $\mu_I = 0.66 \pm 2$ and $\mu_I = 0.75 \pm 0.05$ from the PDF peak for the MOSAiC and the RGPS datasets, respectively. We outline a new method to derive the shape of the yield curve/plastic potential from the shape of the intersection

angles PDF. The resulting shape agrees well with the shape of the teardrop yield curve (Zhang and Rothrock, 2005; Ringeisen et al., 2022), the Mohr–Coulomb curve (Ip et al., 1991), and to a lesser extent, the Curved Diamond (Wang, 2007). We conclude that the popularly used elliptical yield (Hibler, 1979) is not backed by our observations.

Reproducing the observed patterns of LKFs in the Arctic sea ice is one of the remaining challenges of the sea ice modeling community (Hutter et al., 2022). We provide here an observed IAD that can be used as a metric for the evaluation of models,

and we suggest replacing the elliptical yield curve in VP models with a teardrop yield curve for a better representation of simulated LKFs. Such a new setup will need to be tested in high-resolution Arctic simulations to determine if it represents the IAD observed in this study accurately.

*Code and data availability.*    The deformation data based on Sentinel-1 SAR imagery are submitted to PANGAEA (von Albedyll and Hutter, submitted). The RGPS LKF data is available here: Hutter et al. (2019) The LKFs extraction code is available here: Hutter (2019)

The python code to create possible yield curves from intersection angles PDF (and inversely) is available on request and will be made available publicly after the review process.

## Appendix A: Non-conjugate angles

In Sect. 3.4.1, we show that for 37% of the intersection angles (28% for MOSAiC and 38% for RGPS), it is possible to separate the acute and the obtuse angles. However, for many intersecting LKFs, the LKFs' vorticities are the same and it is not possible

to separate between obtuse and acute angles (see Figure 2b). Figure A1 shows the IAD for all angles, conjugate angles, and non-conjugate angles. Note that these PDFs are mirrored relative to 90°, as we cannot differentiate between obtuse and acute angles. For the RGPS dataset, extracting the non-conjugate angles leads to a very uniform IAD: there is still a peak in the IAD, but it is much less dominant. For the MOSAiC dataset, the non-conjugate angles feature many small angles, with peaks around 0° and 180°.

The intersecting LKFs with the same vorticity can have several origins:

1. The time step of the observations is too large to resolve the actual vorticity during the deformation. The vorticity recorded over a (multi-)day period is not necessarily representative of the deformation rates during the formation. Even if two





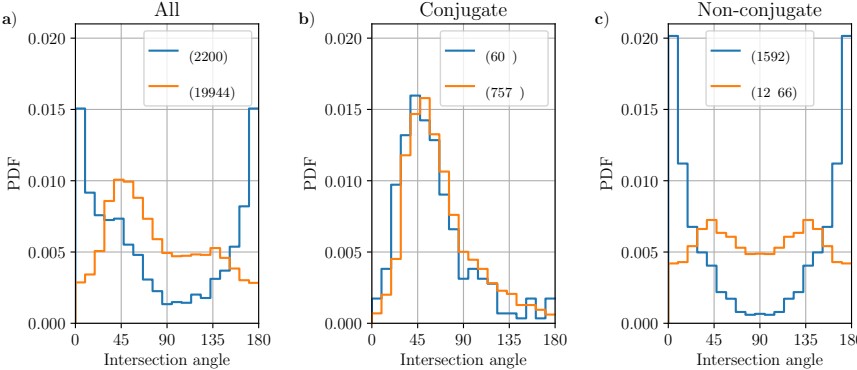

**Figure A1.** Probability Density Function (PDF) of the intersection angles in the Arctic sea ice. The panels show the conjugate and non-conjugate angles for all (left), Conjugate (middle), and non-conjugate (right) angles, as defined on Fig. 2. The numbers in parenthesis show the numbers of intersection angles shown in the PDF.

intersection LKFs are formed under compressive forcing, rapidly changing winds can induce a different ice motion. In this case, the initial failure allows the more mobile ice to deform in shear motion, which leads to the same vorticity sign.

This behavior may be especially present for deformation data with a low temporal resolution, e.g., the RGPS dataset.

2. The presence of the same vorticity sign on both intersecting LKFs could emerge from rotation. LKFs dynamics can involve rotation under shear (Wilchinsky and Feltham, 2004), a process also observed in granular materials (e.g., Oda and Kazama, 1998). This process seems more likely for small-scale observations, e.g., from the MOSAIC Sentinel dataset with a spatial resolution of 1.4 km.

3. If an LKF is not detected properly and cut into two parts, both parts will have the same sign vorticity, and due to their proximity, will be identified as intersecting LFKs in our analysis. We tuned the parameters of the detection algorithms to minimize this effect, but especially for the MOSAiC data, we find instances of this effect.

## Appendix B: Fracture angles from the elliptical yield curve

We inverse the process of Sect. 3.4.2 to extrapolate the intersection angles that would be given by the elliptical curve (yield
curve or plastic potential) in the VP sea ice model. This way, we can compute the expected PDF of the intersection angles in a high-resolution sea ice viscous-plastic model that uses an elliptical yield curve. Figure B1 shows the results of this process for three different aspect ratios $e$. The PDF of the intersection angles for $e = 2$ peaks strongly at 90°. Using a smaller aspect ratio, such as $e = 0.7$ (Bouchat and Tremblay, 2017) flattens the PDF. These PDFs are symmetrical relative to 90° while the observations presented in this paper (Figure 4) are strongly skewed towards small angles. Note that this process to estimate the



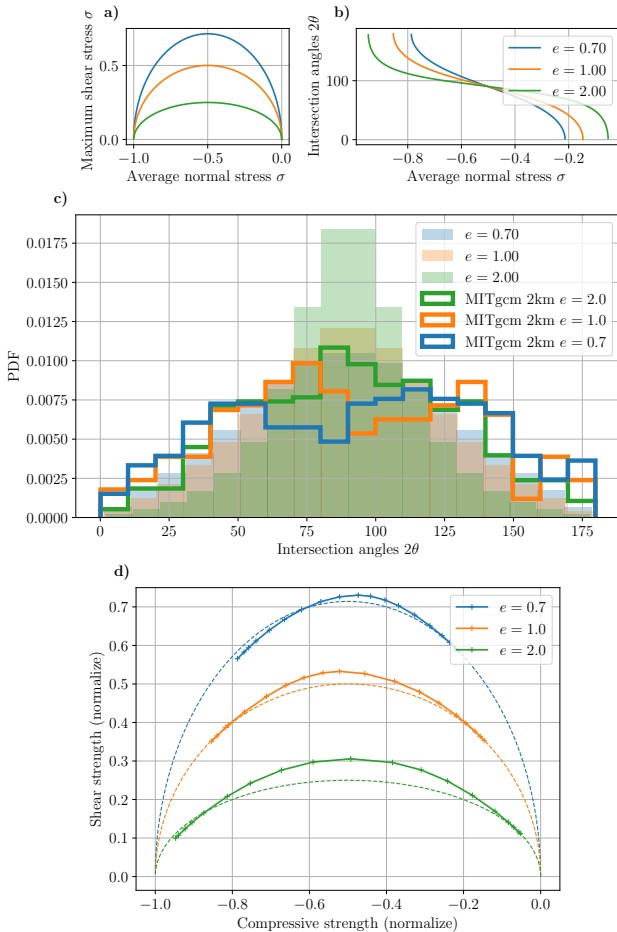

**Figure B1.** Elliptic yield curves in invariant space **(a)**, their theoretical fracture angle as a function of the first invariant $\sigma_I$ **(b)**, the PDF of the theoretical fracture angles, with the PDF of conjugate intersection angles from MITgcm 2 km resolution runs for the same ellipse ratios **(c)**, and the reconstructed yield curve from the modeled IAD **(d)**.

IAD still uses the assumption that all parts of the yield curve are equally probable to be the subject of plastic deformation. An analysis of high-resolution sea ice models would be necessary to see if this hypothesis is valid.

     The histogram lines on Fig. B1c show the PDF of conjugate intersecting angles in a 2 km MITgcm simulation (Hutter et al., 2021) using the same method as described in Sect. 2.2. The histograms of simulated intersection angles show the same evolution of the PDF as expected from theory, with a higher peak around 90° for $e = 2$, and a flatter PDF for $e = 1.0$ and $e = 0.7$. Also,
the PDFs of the models are close to being symmetrical with respect to 90°, despite using the algorithm to separate obtuse and acute angles.

     On Fig. B1d, we use the IAD from the simulations (Fig. B1c) to reconstruct the yield curve for all three values of $e$, as done in Sect. 3.4.2. The resulting yield curve fits well with the elliptical yield curve prescribed in the associated simulations. LKFs



intersection angles can only be created when the slope of the yield curve is within the range $[-1, 1]$. A slope of $-1$ corresponds
to an intersection angle of $0°$, a slope of $+1$ corresponds to an intersection angle of $180°$. To take this into account, we start
reconstructing the yield curve at the point $[\sigma_\mathrm{I}, \sigma_\mathrm{II}]$ where $\frac{\partial \sigma_\mathrm{II}}{\partial \sigma_\mathrm{II}} = -1$, and scale the rest of the yield curve to have the endpoint
at $\sigma_\mathrm{I}$ where the slope is $\frac{\partial \sigma_\mathrm{II}}{\partial \sigma_\mathrm{II}} = 1$.

*Author contributions.* DR coordinated this study, derived the sea ice properties (internal angle of friction and curve shape), made figures, and
wrote sections. NH performed the LKFs extraction, the intersection angles measurement and classifications for both data sets, made figures,
and wrote sections. LvA provided the MOSAiC deformation dataset, made figures, and wrote sections. All authors edited the manuscript.

*Competing interests.* The authors declare that they have no conflict of interest

*Acknowledgements.* The authors would like to thank Jennifer Hutchings, Daniel Watkins, and Bruno Tremblay for their comments on an
earlier version of this work. This project has been supported by the Deutsche Forschungsgemeinschaft (DFG) through the International
Research Training Group "Processes and impacts of climate change in the North Atlantic Ocean and the Canadian Arctic" (grant no. IRTG
1904 ArcTrain).





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
