# Peer review of "Deformation lines in Arctic sea ice: intersection angles distribution and mechanical properties"

_EGUsphere, 2022_

## Referee Comment (RC1)

This paper documents the in-depth analysis of emergent linear deformation features from data sets of sea ice drift and deformation. The analysis is generally well documented and the results well presented. The results presented here will be very interesting to the field of sea ice rheological modelling and recommend it for publication after some minor corrections to make the methodology more understandable and repeatable.

My suggestions for the main corrections to this paper are as follows:

The first is based around figure 3. Can you add more detail, probably in the text, about the distance from the intersection point the intersection angle calculation is based upon. For figure 3a, the angle calculated at the distance of the label appears to be more acute that at a 1/5$^{th}$ of this distance. This is even more pronounced for the 44 degree angle in the non-conjugate case. Is this distance a tuning parameter for the algorithm, can it be tuned, and is it a factor in the difference between the MOSAiC and RGPS data? Have variations to the distance from the intersection point been investigated, and do the final results of this study change with it?

The second is on the topic of reconstructing the yield curve. At the moment it is very difficult to understand exactly how this done from the description in the text. Adding more information is essential to allow this method to be repeatable, and also for the context of the results to be understood. This is true for the results in the main paper body, but even more so for those in appendix B2. I have given more precise comments for this section below, with the detailed minor comments.

L2 This is an awkward sentence as many rheological models don't even explicitly consider LKFs. A statement on emergent deformation features that are linear in nature will be more accurate.

L15 This sentence needs splitting or modifying with an 'or' instead of too many commas.

L18 It is not obvious how shear motion influences the halocline. Can you expand on this as it will be an interesting and relevant inclusion?

L24 Feltham (2005) is another good citation for granular flow.

L23 to L32 This is a good paragraph, but it ends on a note about the method of this paper. Consider splitting this last sentence in order to keep the writing coherent.

L41 Similar to the point above about the abstract, an extra description of what is a LKF in the context of a sea ice model will help.

L44 Keen et al. (2021) is a worthwhile inclusion in this list.

L 50 'phi' has not rendered correctly.

L39 and 55, MDA acronym is only used twice so it will be easier to read with the full term in both cases.

L60 this sentence is a little awkward with the citations next to the question mark, consider moving them to the first mention of the normal flow rule.

L 67 Can you be more explicit what is done beyond the work of Hutter et al. (2019 and 2022)?

L 105-110. This methodology is incredibly difficult to follow, and it is not at all obvious how LKFs are extracted beyond point 1. If this methodology is the same as Hutter et al. (2019), then it does not need to be described in detail. If any modifications have been made, then they need to be described better than is done here. Preferably with another figure showing how it all works.

L 117 a similar point to above, this sentence is filled with jargon and it is not all obvious what is done here. I see that the Hutter et al (2019) data is that which is mainly used, but the modifications made to the algorithms for the MOSAiC data need to documented in a way that is understandable and repeatable.

L 123 This method of calculating angles seem very sensible. Can extra sentence or two be added to explain how the vorticity relates to the principle, or most compressive stress direction? When comparing this description with figure 2, and later distributions, on a first read it seems unclear why there are obtuse examples. More description here will help explain this. For examples figure 2 contains no case for compression two directions, though this case will occur and can cause LKF features (see Heorton et al. 2018 for a model example). I assume the angle technique will account for this case too?

Figure 2, are the red arrows for deformation or stress?

L 140 (Heorton 2018) is relevant here too.

L 155 It is not obvious what Monte Carlo test is performed here to show the accuracy of the distribution. Can a citation for this test be added, or a description of how it is done? Is it in either  (Clauset et al., 2009; Hutter et al., 2019) ?

L 165 A little aside on the method use will help here. Do you plot only the deformation rates from the pixels defined as part of a LKF?

L 169 Can a citation for the deformation regimes in these areas be added?

L 171 Is this reason for more shear speculative? If so can you state that this is an authors hypothesis. A similar issue can arise for the previous point. An extra figure that may help is a averaged map of the LKF deformation rates. This will back up this hypothesis and be a very interesting plot.

L 172 Is it possible to make this comparison when these two data have been normalized as stated previously?

L 176 My interpretation of 5d is that one of the data (I assume MOSAiC due to a previous figure) has a greater number of higher positive dilatancy angles. Is this what you mean?

Figure 5. Are the line colours the same as Figure 4? If so this needs repeating for easier readability. It is not immediately obvious how the normalization is achieved for the 5d. The legends I can see have some formatting and I think the decimal points are missing. This adds to the difficulty interpreting 5d. Will 5d benefit from no normalization?

Equation 4 what is $tau\_0$?

L 208 what is a breaking index and i?

L 221 This functional form seems confusing for me, I understand the concept and it is sound, but should this equation read $sigma\_II = F\_I(sigma\_I)$? Or $F\_I(sigma\_I,sigma\_II) = 0$ (or a constant)? I guess this is a question of how the potential is defined.

L 226 PDF of what? Can you be more explicit on the exact angle data used?

L 228 can you define this origin in numerical coordinates? Is it $sigma\_I$, $sigma\_II = 0$?

L226 – 231 Is this method of curve reconstruction novel? If so it is very useful. Is it possible to include a more theoretical description of what is achieved? As in: using information on which angles to recreate the relation between principle stress components due to which assumptions. Is this done in the previous paragraph? I am currently finding it hard to link the two.

L 227 what assumptions are made for the from of $F\_I$? How is $F\_I$ calculated from the data?
L 228 Which angles, intersection angles or internal friction angles? If intersection angles, why are these associated with the origin or invariant stress?

L 248 citation needs to be in parenthesis.

L 253 'We wonder' is this in relation to the work documented here or a comparison to future work? More defined language is needed.

L 319 The data availability needs to be a working url, not a paper citation. Does one exist from the other publication? If not, then it can not be claimed that the data is freely available.

Appendix B. This section really requires a short introduction. It is described earlier, but repeating the experiments and data used is needed in order to understand it in isolation.

L 345, it is not clear what has been done here. Have the angles been calculated from deformation plots or some other method? If the method in 3.4.2 is reversed then how is

figure B1 d made? As this method seems circular. Or are two methods used for the two pdfs in figure B1 c?

Additionally for figure B1, is the reasoning behind this appendix to show that the method can reconstruct a yield curve from deformation along intersection angles when using a model? If so then figure B1 is compelling, though can you comment on the larger difference for the e=2.0 case, and what context this has for the results in the main paper body?

[1]
Feltham, D.L. 2005. Granular flow in the marginal ice zone. *Philosophical Transactions of the Royal Society A: Mathematical, Physical and Engineering Sciences*. 363, 1832 (Jul. 2005), 1677–1700. DOI:https://doi.org/10.1098/rsta.2005.1601.

[1]
Keen, A. et al. 2021. An inter-comparison of the mass budget of the Arctic sea ice in CMIP6 models. *The Cryosphere*. 15, 2 (Feb. 2021), 951–982. DOI:https://doi.org/10.5194/tc-15-951-2021.

[1]
Heorton, H.D.B.S. et al. 2018. Stress and deformation characteristics of sea ice in a high-resolution, anisotropic sea ice model. *Phil. Trans. R. Soc. A*. 376, 2129 (Sep. 2018), 20170349. DOI:https://doi.org/10.1098/rsta.2017.0349.

---

## Author Comment (AC1)

**Answers to reviewers - egusphere-2022-1481**

Damien Ringeisen, Nils Hutter, Luisa von Albedyll,

Correspondence: damien.ringeisen@mcgill.ca, nhutter@uw.edu

May 4, 2023

Dear Editor,

We thank the two reviewers for their comments and advice that helped to strengthen and clarify our manuscript. In the following pages, we answer each comment and suggestion of the reviewers.

Yours sincerely,
Damien Ringeisen, Nils Hutter, and Luisa von Albedyll

**Note:**

- The reviewers' comments are shown in black.

- The authors' answers are shown in blue.

- **The modifications of the manuscript are shown in bold blue.**

- The line numbers refer to the updated manuscript.

**1    Reviewer 1 - Harry Heorton**

- **R1#1,** This paper documents the in-depth analysis of emergent linear deformation features from data sets of sea ice drift and deformation. The analysis is generally well documented and the results well presented. The results presented here will be very interesting to the field of sea ice rheological modelling and recommend it for publication after some minor corrections to make the methodology more understandable and repeatable. My suggestions for the main corrections to this paper are as follows:

  We thank the reviewer for their comments and suggestions that made the paper clearer and stronger.

- **R1#2,** The first is based around figure 3. Can you add more detail, probably in the text, about the distance from the intersection point the intersection angle calculation is based upon. For figure 3a, the angle calculated at the distance of the label appears to be more acute that at a 1/5th of this distance. This is even more pronounced for the 44 degree angle in the non-conjugate case. Is this distance a tuning parameter for the algorithm, can it be tuned, and is it a factor in the difference between the MOSAiC and RGPS data? Have variations to the distance from the intersection point been investigated, and do the final results of this study change with it?

  Up to 10 points away from the intersection point on each side are used to compute the intersection angle. If the distance between the intersecting point and the end of the LKF is less than 10 pixels, all pixels in between are used to compute the orientation of the LKF. We use the value of 10 to suppress the effect of discrete orientations for computed over shorter segments of the LKF caused by the pixel structure and the morphological thinning. In summary, depending on the length of the LKF 21 pixels down to 7 pixels (minimum length of detected LKFs) are used to compute the orientations of the LKFs.

  **In the method section, L139, we add the following sentence "*The angles are measured from points ca. 10 points away (in both directions) from the intersection point, to avoid the effects of discrete orientations on a grid. In practice, because some LKFs are shorter, the number varies from 7 to 21 points. The number of 10 points was chosen to be a good compromise to get an accurate result and avoid discrete effects.*"**

- **R1#3,** The second is on the topic of reconstructing the yield curve. At the moment it is very difficult to understand exactly how this done from the description in the text. Adding more information is essential to allow this method to be repeatable, and also for the context of the results to be understood. This is true for the results in the main paper body, but even more so for those in appendix B2. I have given more precise comments for this section below, with the detailed minor comments.

  We have modified and expanded this section considering your minor comments to make sure that the method is clearer to the reader. Thank you for your help with the minor comments.

  **We add a new figure detailing the process, see Fig. 1**

**1.1    Detailed comments**

- **R1#4,** L2 This is an awkward sentence as many rheological models don't even explicitly consider LKFs. A statement on emergent deformation features that are linear in nature

[Figure]

Figure 1: Approximation of the yield curve or plastic potential from the intersection angles distribution (IAD). Each bin of the distribution is used to create a segment of the curve with the length of the PDF value and the angle corresponding to the center of the bin, starting from the smaller bin.

will be more accurate.

Agreed, corrected as suggested.

**The sentence on L4 will now read "*The LKFs emerging from sea ice rheological models do not reproduce the observed LKFs intersection angles.*"**

- **R1#5,** L15 This sentence needs splitting or modifying with an 'or' instead of too many commas.

  Corrected as suggested

- **R1#6,** L18 It is not obvious how shear motion influences the halocline. Can you expand on this as it will be an interesting and relevant inclusion?

  We added some details.

  **The sentence on L23 will now read "*Second, shear motion and sea-ice growth along LKFs influence the halocline through pycnocline upwelling and brine injection, respectively...*"**

- **R1#7,** L24 Feltham (2005) is another good citation for granular flow.

  Corrected as suggested

- **R1#8,** L23 to L32 This is a good paragraph, but it ends on a note about the method of this paper. Consider splitting this last sentence in order to keep the writing coherent.

Effectively this sentence was not coherent, we will rewrite it.

**The last sentence on L36-37 will now read:** "*To overcome those shortcomings and still retrieve the mechanical properties from the orientation of the failure lines, the vorticity at the intersections of LKFs can be used instead. By describing the rotation of ice during deformation, vorticity can be utilized to infer the main stress direction and eventually link it to the intersection angles of the LKFs (Hutter et al., 2022).*"

- **R1#9,** L41 Similar to the point above about the abstract, an extra description of what is a LKF in the context of a sea ice model will help.

  Agreed

  **We will add this sentence in L49 to make this clearer:** "*LKFs in sea ice models emerge from the rheological model, especially from the threshold mechanism of some properties. This threshold mechanism creates LKFs because it includes a change of mechanical properties between large and plastic deformations, in the LKFs, and small deformations, in between the LKFs, i.e., viscosity maximum for VP models (Hutchings et al., 2005) and damage for brittle models (Dansereau et al., 2016).*"

- **R1#10,** L44 Keen et al. (2021) is a worthwhile inclusion in this list.

  Added as suggested

- **R1#11,** L50 'phi' has not rendered correctly.

  Corrected as suggested

- **R1#12,** L39 and 55, MDA acronym is only used twice so it will be easier to read with the full term in both cases.

  Corrected as suggested

- **R1#13,** L60 this sentence is a little awkward with the citations next to the question mark, consider moving them to the first mention of the normal flow rule.

  Corrected as suggested

- **R1#14,** L 67 Can you be more explicit what is done beyond the work of Hutter et al. (2019 and 2022)?

  We use the same approach to select conjugate LKFs as described in Hutter et al. (2022). We have made this more explicit in the text and rephrased:

  **L78:** "*Second, following the approach from (Hutter et al., 2022), we consider only conjugate pairs of LKFs, i.e., intersecting LKFs that formed simultaneously under compressive forcing.*"

- **R1#15,** L 105-110. This methodology is incredibly difficult to follow, and it is not at all obvious how LKFs are extracted beyond point 1. If this methodology is the same as Hutter et al. (2019), then it does not need to be described in detail. If any modifications have been made, then they need to be described better than is done here. Preferably with another figure showing how it all works.

We used the same methods described in Hutter et al. (2019). We intended to provide a short summary of the described methods with this paragraph. We believe that our summary is a worthy contribution to our paper for readers that are generally familiar with the topic and/or have read the publication but might not recall all details. That's why we would like to keep the paragraph. We have clarified in the text that the approach is identical to Hutter et al. (2019).

**We have added the sentence on L119:** "*Here, we provide a short summary of the algorithms and direct the interested reader to the details in Hutter et al. (2019).*"

- **R1#16,** L 117 a similar point to above, this sentence is filled with jargon and it is not all obvious what is done here. I see that the Hutter et al. (2019) data is that which is mainly used, but the modifications made to the algorithms for the MOSAiC data need to documented in a way that is understandable and repeatable.

  We added some details to the description to clarify the modification made. We prefer not to go into depth in this method part, as we regard the modifications as minor and are afraid to distract the reader from the main topic of the paper. We will, however, release a new version of the `lkf_tools` on zenodo and github once the paper is accepted. We refer the interested reader than to the documentation of the newly released code.

  **L130ff:** "*(1) We apply a directional filter to the input deformation rates to reduce grid-scale noise. The directional filter is a 1-d kernel spanning 7 pixels that is rotated at each pixel over all directions to compute the variability along different directions. We choose the direction of lowest variability to apply the 1-d filter and compute the filtered deformation rates. This allows us to reduce noise but still preserve the linear structure of LKFs in the deformation data. (2) The morphological thinning routine was modified to align the LKF skeletons in the binary maps to the position of the highest deformation rates across the LKF. Details to both modification can be found in the routines in `dir_filter.py` in the newly released version of the code (Hutter, 2023).*"

- **R1#17,** L 123 This method of calculating angles seem very sensible. Can extra sentence or two be added to explain how the vorticity relates to the principle, or most compressive stress direction? When comparing this description with figure 2, and later distributions, on a first read it seems unclear why there are obtuse examples. More description here will help explain this. For examples figure 2 contains no case for compression two directions, though this case will occur and can cause LKF features (see Heorton et al. 2018 for a model example). I assume the angle technique will account for this case too?

  We consider here the principal axes of stress. We can have LKFs form in the case of bi-axial compression, but then it means that one stress is higher than the other, hence the maximum shear stress (2nd invariant) $\sigma_{II} \neq 0$ and the principal stress $\sigma_1 > \sigma_2$. and this is equivalent to having a uniaxial compression case. In the case where $\sigma_1 = \sigma_2$ it is a uniform compression case, and no LKFs should form. We add a sentence to clarify this.

  **The sentence on L142-146 of the manuscript now reads** "*For conjugate angles, the principal stress direction can be identified from the resulting ice motion. The ice flows from the most compressive to the least compressive principal stress. Note, by convention, as compression is negative, the first*"

*principal stress direction is the direction with the lower compressive stress, while the second is the higher compressive stress. We do not consider the exact stresses here but only the principal stresses and their direction, therefore it includes also bidirectional compression situations.*"

- **R1#18,** Figure 2, are the red arrows for deformation or stress?

  The red arrow are the stresses.

  **The last sentence of the caption of Figure 2 now reads** "*The vorticity of the drift allows knowing the direction of the principal stresses (red arrows), hence making the difference between conjugate and non-conjugate failure lines.*"

- **R1#19,** L 140 (Heorton 2018) is relevant here too.

  Added as suggested

- **R1#20,** L 155 It is not obvious what Monte Carlo test is performed here to show the accuracy of the distribution. Can a citation for this test be added, or a description of how it is done? Is it in either (Clauset et al., 2009; Hutter et al., 2019) ?

  We clarify this sentence.

  **The sentence on L178 of the manuscript now reads** "*The goodness of the fit is tested with a Monte-Carlo test with 10 000 different random sub-samples, taking into account the discrete nature of intersection angles between LKFs that are defined on a regular grid (Hutter et al., 2019), and using a Kolmogorov–Smirnov (KS) statistic (see details in Clauset et al., 2009; Hutter et al., 2019).*"

- **R1#21,** L 165 A little aside on the method used will help here. Do you plot only the deformation rates from the pixels defined as part of a LKF?

  Yes. For the distributions shown in Fig. 5, we only use deformation rates from the pixels defined as part of a LKF. We have clarified this in the text and added a sentence.

  **On L189** "*We extract the deformation rates from all pixels defined as part of a LKF to compare the characteristics of the deformation in the MOSAiC and RGPS dataset. Like for the IAD, both the MOSAiC and RGPS dataset agree in shape ...*"

- **R1#22,** L 169 Can a citation for the deformation regimes in these areas be added?

  We have rephrased this sentence to reflect that it is our understanding of the deformation regimes.

  **We have modified the sentence on L194 to:** "*We speculate that this reflects a generally more divergent regime in the Transpolar drift compared to the Beaufort Gyre which features more compressive settings* "

- **R1#23,** L 171 Is this reason for more shear speculative? If so can you state that this is an authors hypothesis. A similar issue can arise for the previous point. An extra figure

that may help is a averaged map of the LKF deformation rates. This will back up this hypothesis and be a very interesting plot.

Thanks for pointing this out. We have made clear that we are speculating. We agree that such an extra figure would be interesting but we think it would lead to an analysis beyond the scope of our paper on the intersection angles. However, as both datasets are now publicly available we invite and encourage our peers to do such an analysis.

**We have added a few words on L194ff:** "*We speculate that this reflects a generally more divergent regime in the Transpolar drift compared to the Beaufort Gyre which features more compressive settings. The RGPS data set shows higher shear deformation that could potentially originate from the circular motion of the Beaufort Gyre. The higher shear rates also result in higher total deformation rates in the RGPS data set.*"

- **R1#24,** L 172 Is it possible to make this comparison when these two data have been normalized as stated previously?

  We guess that the reviewer is here referring the dilatancy angles. The dilatancy angles are computed before normalizing the data. We now specify this fact.

  **We add on L199 this sentence** "*Here, the divergence and shear are used before normalization.*"

- **R1#25,** L 176 My interpretation of 5d is that one of the data (I assume MOSAiC due to a previous figure) has a greater number of higher positive dilatancy angles. Is this what you mean? Figure 5. Are the line colours the same as Figure 4? If so this needs repeating for easier readability. It is not immediately obvious how the normalization is achieved for the 5d. The legends I can see have some formatting and I think the decimal points are missing. This adds to the difficulty interpreting 5d. Will 5d benefit from no normalization?

  We have noticed that the formatting of this figure went wrong during the online publication. This includes repeating the line colors. There is no normalization for 5d, we corrected the figure (see above) and described this in the text.

- **R1#26,** Equation 4 what is $\tau_0$?

  The cohesion, i.e., the shear strength when no compression nor tension are present. Corrected.

- **R1#27,** L 208 what is a breaking index and i?

  It is a concept developed by (Erlingsson, 1991). We have to admit that we have trouble understanding the details of the concept. We have moved the part about Erlingsson's work to the discussion and have given some context to it.

  **We extended the discussion with the following sentence on L284** "*In contrast, our estimates disagree with the findings of Erlingsson (1988, 1991). Applying a breaking index of $i = 2$ defined by his methodological framework, the estimated internal angles of friction are $\phi \simeq 25°$ or $\mu \simeq 0.13$ and $\phi \simeq 65°$ or $\mu \simeq 0.82$. However, since we find that this framework does not agree with the creation of LKFs in sea ice models, we rather focus on the Mohr−Coulomb's framework.* "

- **R1#28,** L 221 This functional form seems confusing for me, I understand the concept and it is sound, but should this equation read $\sigma_{\mathrm{II}} = F_{\mathrm{I}}(\sigma_{\mathrm{I}})$? Or $F_{\mathrm{I}}(\sigma_{\mathrm{I}}, \sigma_{\mathrm{II}}) = 0$ (or a constant)? I guess this is a question of how the potential is defined.

  The former equation you mentioned ($\sigma_{\mathrm{II}} = F_{\mathrm{I}}(\sigma_{\mathrm{I}})$) is the correct version, but we rewrite this part because it would be confusing. The potential used for the derivation of the constitutive equations (Ringeisen et al., 2021) would be $F(\sigma_{\mathrm{I}}, \sigma_{\mathrm{II}}) = \sigma_{\mathrm{II}} - \sigma_{\mathrm{II}}(\sigma_{\mathrm{I}})$.

- **R1#29,** L 226 PDF of what? Can you be more explicit on the exact angle data used?

  Agreed.

  **We added for each bin of angles in the observed PDF "*of the intersection angles (Figure 4)*"**

- **R1#30,** L 228 can you define this origin in numerical coordinates? Is it $\sigma_{\mathrm{I}}$, $\sigma_{\mathrm{II}} = 0$?

  Yes, it is.

  **We add on L254 "$(\sigma_{\mathrm{I}}, \sigma_{\mathrm{II}}) = (0, 0)$"**

- **R1#31,** L226 – 231 Is this method of curve reconstruction novel? If so it is very useful. Is it possible to include a more theoretical description of what is achieved? As in: using information on which angles to recreate the relation between principle stress components due to which assumptions. Is this done in the previous paragraph? I am currently finding it hard to link the two.

  Yes it is quite novel. A similar method was used by (Wang, 2007) (See the curved diamond on Fig. 8) but they did not use the method to discriminate between acute and obtuse angles and therefore did not use a distribution.

  **For a more intuitive understanding of our method, we added a figure illustrating the method, see Fig 1 above.**

- **R1#32,** L 227 what assumptions are made for the form of $F_{\mathrm{I}}$? How is $F_{\mathrm{I}}$ calculated from the data?

  We do not compute $F_{\mathrm{I}}$ as such, it is only a succession of lines that create a shape of the yield curve. Each line is defined using the Mohr–Coulomb criterion.

- **R1#33,** L 228 Which angles, intersection angles or internal friction angles? If intersection angles, why are these associated with the origin or invariant stress?

  It is the intersection angle in this case. We now specify this in the text. Also, the new figure (Fig 1 above) will, we think, clarify the process.

  **L254 of the manuscript now reads "*We start from the smallest intersection angles of the distribution, as they are linked to the steepest curve slopes from Eq. (7), and iterate through the PDF, with the start point of each segment being the tip of the previous segment.*"**

- **R1#34,** L 248 citation needs to be in parenthesis.

  Changed as suggested

- **R1#35,** L 253 'We wonder' is this in relation to the work documented here or a comparison to future work? More defined language is needed.

  Those thoughts go indeed beyond the content of this manuscript. We have changed this. We now write:

  **We changed the sentence on L278 to "*Future work should investigate how the shape of...*"**

- **R1#36,** L 319 The data availability needs to be a working url, not a paper citation. Does one exist from the other publication? If not, then it can not be claimed that the data is freely available.

  We have updated the section and added urls.

  **The new text is now (L349) " *The deformation data based on Sentinel-1 SAR imagery is made available on PANGAEA under the identifier: PDI-33705 (von Albedyll and Hutter, tted). The RGPS LKF data is available available on PANGAEA: https://doi.org/10.1594/PANGAEA.898114 Hutter et al. (2019)*"**

- **R1#37,** Appendix B. This section really requires a short introduction. It is described earlier, but repeating the experiments and data used is needed in order to understand it in isolation.

  We add a short introduction,

  **We add on L376 "*In this appendix, we show, as a proof of concept, that the method used in Section 3.4.2 allows to reconstruct the shape of the elliptical yield curve from the model deformation output. First, we compute the theoretical intersection angles distribution from the yield curve shape. Then, we extract the intersection angles distribution from 2-km resolution Pan-Arctic simulations. Finally, we reconstruct the yield curve shape from the modeled intersection angles distribution and show that it gives a shape very similar to the yield curve of the first step.*"**

- **R1#38,** L 345, it is not clear what has been done here. Have the angles been calculated from deformation plots or some other method? If the method in 3.4.2 is reversed then how is figure B1 d made? As this method seems circular. Or are two methods used for the two pdfs in figure B1 c

  The shaded area is the theoretical PDF from the yield curve shape. The lines are the PDF from modeled intersection angles. Yes, the angles are also extracted from deformation, but from deformation fields of a model. We clarify all this in the new appendix.

  **The appendix was extensively rewritten.**

- **R1#39,** Additionally for figure B1, is the reasoning behind this appendix to show that the method can reconstruct a yield curve from deformation along intersection angles when using a model? If so then figure B1 is compelling, though can you comment on the larger difference for the $e = 2.0$ case, and what context this has for the results in the main paper body?

Yes, we show that the method can reconstruct the yield curve of a model from simulated deformations. We add a sentence clarifying the goal of this appendix. We also add a sentence discussing the larger difference with $e = 2.0$

We add on L396 " *There is a larger difference for $e = 2$ case, intersection angles around $90°$ are less present than we would expect from the model. Globally, we see that all three PDFs of the modeled intersection angles are missing angles around $90°$. This could be the result of a small departure from the hypothesis that all parts of the yield curve are equally probable.*"

**1.2 Reviewer's bibliography**

Feltham, D.L. 2005. Granular flow in the marginal ice zone. Philosophical Transactions of the Royal Society A: Mathematical, Physical and Engineering Sciences. 363, 1832 (Jul. 2005), 1677–1700. DOI: `https://doi.org/10.1098/rsta.2005.1601`.

Keen, A. et al. 2021. An inter-comparison of the mass budget of the Arctic sea ice in CMIP6 models. The Cryosphere. 15, 2 (Feb. 2021), 951–982. DOI: `https://doi.org/10.5194/tc-15-951-2021`.

Heorton, H.D.B.S. et al. 2018. Stress and deformation characteristics of sea ice in a high-resolution, anisotropic sea ice model. Phil. Trans. R. Soc. A. 376, 2129 (Sep. 2018), 20170349. DOI: `https://doi.org/10.1098/rsta.2017.0349`.

**2 Reviewer 2 - Anonymous**

**R2#1,** The paper fills an important gap in the knowledge of LKF characteristics. I expect it to have an immediate impact on the modelling of sea ice rheology. It has a rigorous background, which leads to it being a bit technical and difficult to read in some places. A reread with a less expert reader in mind would beneficial.

Thank you for your answer. We have added more details, especially in the introduction, abstract, and methods section, to provide background information to a less expert reader. The track changes highlight our additions. However, we are aware that parts of the paper are still addressed to expert readers. We made sure that all important references are included to ensure beginners can read up on the relevant theory.

**2.1 Specific comments**

- **R2#2,** On figure 1 RGPS is blue and MOSAiC red while on the other figures you use orange and blue respectively. Please make the colours consistent to avoid unnecessary confusion.

  Thanks for pointing out this inconsistency. We have updated the colors in Figure 1.

- **R2#3,** Line 110: You say you want to exclude the influence of the coast but on the figure the blue area almost touches the coastline. Can you explain how much should be included and why more precisely? Hutter et al. 2019 commented on the influence of the coastline on LKF distribution further away than this, can you rule out an influence on the angles?

  RGPS observations along the coast are sparser than for example in the Beaufort Sea. Also, the 10-km grid of RGPS extends only up to 100 km close to the coast, beyond which only a 25-km grid is used, which will result in on average lower deformation rates. To detect LKF pixels we use a Difference of Gaussian Filter with kernel sizes of 62.5 km. All these points make it very unlikely that LKFs pixels are detected within 100 km to the coast and even less likely that an intersecting point between two LKFs lies within this range. We do not use an explicit filter for LKF pixels too close to the coast, but even if present the amount of coastal LKF is so small compared to the intersecting LKFs in the Arctic Ocean that we do not expect a significant impact on the presented intersection angle PDFs.

  **We have updated Figure 1 (overview map) in such a way that increasing opacity indicates data coverage. This way it becomes clearer that the majority of the RGPS data is collected far away from the coast.**

- **R2#4,** Line 208: This requires more explanation on what is wrong with the Erlingsson framework

  We admit that we have trouble understanding Erlingsson's framework in full depth. We have moved the comparison of our findings with his results to the discussion and have given some more context why we concentrate on Mohr-Coulomb's framework.

  **L284 We added: "*In contrast, our estimates disagree with the findings of Erlingsson (1988, 1991). Applying a breaking index of $i = 2$ defined by his methodological framework, the estimated internal angles of friction are $\phi \simeq 25°$ or $\mu \simeq 0.13$ and $\phi \simeq 65°$ or $\mu \simeq 0.82$. However, since we find that this framework does not agree with the creation of LKFs in sea ice models, we***

*rather focus on the Mohr–Coulomb's framework. "*

- **R2#5,** Line 328: The large number of small non-conjugate angles in the MOSAiC dataset really stands out. Is the problem with LKFs being cut in two parts you mention sufficient to explain the large difference with RGPS or might there be an effect of the resolution or some other difference?

  We are confident that the LKFs cut into two or more parts are the main cause for these differences. One needs to keep in mind that by cutting one LKF into smaller parts also the number of "false" intersecting points and angles originating from these is higher than the one intersecting point of two longer LKFs, which intersect. This amplifies the effect and leads to these high amounts of non-conjugate angles.

**2.2   small mistakes**

- **R2#6,** Line 169: figure 5a instead of 5b

  Corrected as suggested

- **R2#7,** Line 308: typo in 0.66±2

  Corrected as suggested

- **R2#8,** Line 240: the reference is wrong

  Thanks for spotting this. We have corrected the reference.

  **Now it states: Hutter et al. (2022)**

- **R2#9,** Line 241 and following: has two versions of the same reference

  Thanks for pointing this out. We have removed the reference to the open archive article with the revised manuscript. Now there is only Hutter et al. (2022)

- **R2#10,** Line 248: typo in the reference

  Did you refer to the lack of a bracket? We have added a bracket.

- **R2#11,** Line 477: correct the doi

  Corrected as suggested

**References**

Clauset, A., Shalizi, C. R., and Newman, M. E. J. (2009). Power-Law Distributions in Empirical Data. *SIAM Review*, 51(4):661–703. Publisher: Society for Industrial and Applied Mathematics.

Dansereau, V., Weiss, J., Saramito, P., and Lattes, P. (2016). A Maxwell elasto-brittle rheology for sea ice modelling. *The Cryosphere*, 10(3):1339–1359.

Erlingsson, B. (1988). Two-dimensional deformation patterns in sea ice. *Journal of Glaciology*, 34(118):301–308.

Erlingsson, B. (1991). The propagation of characteristics in sea-ice deformation fields. *Annals of Glaciology*, 15(1):73–80.

Hutchings, J. K., Heil, P., and Hibler, W. D. (2005). Modeling Linear Kinematic Features in Sea Ice. *Monthly Weather Review*, 133(12):3481–3497.

Hutter, N., Bouchat, A., Dupont, F., Dukhovskoy, D., Koldunov, N., Lee, Y. J., Lemieux, J.-F., Lique, C., Losch, M., Maslowski, W., Myers, P. G., Ólason, E., Rampal, P., Rasmussen, T., Talandier, C., Tremblay, B., and Wang, Q. (2022). Sea Ice Rheology Experiment (SIREx): 2. Evaluating Linear Kinematic Features in High-Resolution Sea Ice Simulations. *Journal of Geophysical Research: Oceans*, 127(4):e2021JC017666. _eprint: https://onlinelibrary.wiley.com/doi/pdf/10.1029/2021JC017666.

Hutter, N., Zampieri, L., and Losch, M. (2019). Leads and ridges in Arctic sea ice from RGPS data and a new tracking algorithm. *The Cryosphere*, 13(2):627–645. Publisher: Copernicus GmbH.

Hutter, N., Zampieri, L., and Losch, M. (2019). Linear Kinematic Features (leads & pressure ridges) detected and tracked in RADARSAT Geophysical Processor System (RGPS) sea-ice deformation data from 1997 to 2008.

Ringeisen, D., Tremblay, L. B., and Losch, M. (2021). Non-normal flow rules affect fracture angles in sea ice viscous–plastic rheologies. *The Cryosphere*, 15(6):2873–2888. Publisher: Copernicus GmbH.

von Albedyll, L. and Hutter, N. (submitted). High resolution sea ice drift and deformation in the transpolar drift during mosaic 2019/2020. PANGAEA.

Wang, K. (2007). Observing the yield curve of compacted pack ice. *Journal of Geophysical Research: Oceans*, 112(C5):C05015.